# Immunogenic Cross-Reactivity between Goose and Muscovy Duck Parvoviruses: Evaluation of Cross-Protection Provided by Mono- or Bivalent Vaccine

**DOI:** 10.3390/vaccines10081255

**Published:** 2022-08-04

**Authors:** Vilmos Palya, Anna Zolnai, Balázs Felföldi

**Affiliations:** 1Scientific Support and Investigation Unit, Ceva-Phylaxia Co. Ltd., Szallas utca 5, 1107 Budapest, Hungary; 2Ceva Animal Health, 8906 Rosehill Road, Lenexa, KS 66215, USA

**Keywords:** waterfowl parvoviruses, goose and muscovy duck parvovirus, cross-immunity, inactivated vaccine, vaccination

## Abstract

To investigate the immunogenic cross reactivity between goose parvovirus (GPV) and Muscovy duck parvovirus (MDPV), cross-neutralization was carried out with serum samples collected from birds after infection with one of the two waterfowl parvoviruses. The significantly higher virus neutralization titer obtained against the homologous virus than against the heterologous one suggests important differences between the GPV and MDPV antigenic make up that affects the induced protective virus-neutralizing antibody specificity. This was further confirmed by cross-protection studies carried out in waterfowl parvovirus antibody-free Muscovy ducks immunized at one day of age with whole-virus inactivated oil-emulsion vaccines containing either GPV or MDPV as a monovalent vaccine, or both viruses as a bivalent vaccine. Protection against the clinical disease (growth retardation and feathering disorders) provided by the monovalent vaccine was complete against homologous virus challenge at 2 weeks post-vaccination, while the protection against the heterologous virus challenge was significantly lower (*p* < 0.001). Only the bivalent vaccine containing both goose and Muscovy duck parvoviruses in an inactivated form protected the birds (90–100%) against both waterfowl parvoviruses that can cause disease in Muscovy ducks. Both the cross-neutralization and cross-protection results indicated that adequate protection in Muscovy ducks against the two waterfowl parvoviruses could be achieved only with a vaccine containing both goose and Muscovy duck parvoviruses. Our results showed that the inactivated vaccine applied at one day of age could induce fast immunity (by 2 weeks post-vaccination), providing complete clinical protection in maternal antibody-free birds. It was also demonstrated that day-old vaccination of ducks with maternal antibodies with bivalent vaccine induced active immunity, resulting in 90 to 100% protection by 3 weeks of age, after the decline of maternal antibodies. A booster vaccination administered at 3 weeks of age following the day-old vaccination resulted in a strong and durable immunity against the clinical disease during the susceptible age of the birds.

## 1. Introduction

Waterfowl parvoviruses (WPVs) are members of the Dependoparvovirus genus of the Parvoviridae family, and all of them belong to the same Anseriform dependoparvovirus 1 species [1]. Phylogenetic analysis has shown that WPVs can be divided into two major genetic groups according to their host specificity, namely, goose parvovirus (GPV) and Muscovy duck parvovirus (MDPV) [2,3]. In recent years, a new disease condition known as short beak and dwarfism syndrome (SBDS) or beak atrophy and dwarfism syndrome (BADS) has been described in mule ducks and Cherry Valley ducks [4,5]. It has been demonstrated that the virus causing SBDS/BADS is a distinct lineage of goose parvovirus [4], named by Chinese scientists as a novel goose parvovirus [6].

The genome of WPV contain two open reading frames that encode three capsid proteins (VP1, VP2, and VP3), and two nonstructural proteins (NS1 and NS2). The C-terminal portion of the VP1 gene contains the coding sequences of VP2 and VP3, which are expressed via differential splicing [2]. Genomic sequence analysis of GPV, MDPV and SBDS viruses revealed that the VP1 polypeptides of GPV and MDPV share a nucleotide similarity of 79.6–85.5% at the genome level [2,7], while SBDS virus isolates had 95.1–98.2% identity with classical GPV and 88.0–92.6% identity with MDPV which suggests that there may be a closer immunogenic cross reactivity of SBDS virus with GPV than with MDPV [6,8]. The nucleotide differences of VP1 between GPVs and MDPVs are about 20–24%, while within the GPV and MDPV groups the differences are only about 0.1–7.0 and 0.1–1.9%, respectively [7,8]. VP3 is the most variable and abundant of the three core proteins, and is responsible for the induction of neutralizing antibodies that confer protective immunity [8,9,10].

Waterfowl parvoviruses cause the most dreadful disease of goslings and Muscovy ducklings [3,11,12,13]. Occasionally the disease accounts for mortality of 70 to 100% in susceptible flocks when breeders transfer the infection vertically to the progenies or the infection occurs at an early age of life. GPV and MDPV differ in the host range; while geese are fully resistant to MDPV infection, in Muscovy ducks both viruses can cause severe disease [3,14]. In addition, SBDS/BADS, first reported in the 1970s in mule ducks (crossbreed of Pekin duck and Muscovy duck) and recently in Cherry Valley ducks, can cause also significant economic losses in affected flocks, due to strong growth retardation of the animals [4,15].

The disease caused by WPVs is strictly age dependent. In susceptible goslings and ducklings less than 1 week of age, 100% mortality may occur, while the losses above this age decrease with the age [3,13,15]. In birds with an impaired immune system the infection may cause significant economic losses up to 6 to 8 weeks of age [3,16]. Breeder geese and Muscovy ducks that have been naturally infected or vaccinated transfer maternal antibodies via the egg yolk to their progeny. This passively acquired antibody may persist until 2 to 6 weeks of age depending on the day-old antibody levels of individual birds [3]. Since the disease is confined to young geese and Muscovy ducks, control measures have been aimed at providing adequate immunity during the first 6 to 8 weeks of life [3]. To achieve this, different methods have been applied during the last four decades. These include: (i) passive immunization of newly hatched birds with convalescent or hyperimmune serum or (ii) active immunization of both young and adult animals either with attenuated or inactivated vaccine alone, or in the combination of the two vaccines [3,17,18,19,20]. Attenuated vaccines can confer fast and strong protection in young animals but only when it is given to birds with no or very low levels of maternally derived antibodies (MDA) to GPV and/or MDPV [3,14,17,21]. On the other hand, inactivated vaccines induce a slower immune response, but are less sensitive to the interference with maternal antibodies than live vaccines [10,19,21].

Considering the identified genetic differences between GPV and MDPV and field experiences with vaccination it has been suggested that only bivalent vaccines containing both goose and MDPV antigens could ensure a high level of clinical protection against the two waterfowl parvoviruses causing disease in Muscovy ducks. However, it has not yet been investigated what is the level of cross-protection afforded by an inactivated monovalent GPV or MDPV vaccine against challenge with a heterologous virus when compared to a bivalent (containing both goose and Muscovy duck parvoviruses) vaccine. To investigate this, monovalent GPV and MDPV inactivated vaccines were prepared and compared with a commercially available bivalent vaccine (containing both GPV and MDPV antigens) by testing their immunogenicity in Muscovy ducklings, which were either free from or carrying maternally derived antibodies to both GPV and MDPV.

## 2. Materials and Methods

### 2.1. Animals and Housing

Day-old MDA-free Muscovy ducklings, hatched from a breeder flock free from GPV and MDPV, were purchased from ANSES Ploufragan Laboratories (22440 Ploufragan, France). Conventional MDA-positive day-old Muscovy ducklings, hatched from a breeder flock regularly vaccinated against parvoviruses, were purchased from Prophyl Kft. (Mohacs, Hungary). On the day following hatching they were allocated to treatment groups by randomization according to their body weight and were housed in separate, isolated rooms in contained animal facilities. They were kept on the floor with litter and fed with high-quality commercial duck rations with unlimited access to water. The ducklings were identified by individually numbered leg rings.

All animal studies were conducted in compliance with the applicable animal welfare regulation in force at the location and time of the experiments (Hungarian Act No. XXVIII/1998 and BA01/2005). The humane endpoint was determined as the phase when severely diseased ducks were not able to eat or drink any more (euthanized ducks were counted as mortality on the day of euthanasia). Ducks reaching the humane endpoint and all surviving birds at the end of the post-challenge observation period were euthanized by intracardiac injection of sodium pentobarbital (Euthoxin, Alpha-Vet Ltd., Budapest, Hungary).

### 2.2. Vaccines

The inactivated vaccines used in the experiments were either monovalent or bivalent preparations. The vaccine preparations used in the experiments contained whole-virus antigens of GPV or MDPV or both, inactivated by binary ethylenimine and formulated as a water-in-oil emulsion. To evaluate the extent of cross-protection provided by any of the two inactivated whole parvovirus antigens (GPV or MDPV) against challenges with the homologous or heterologous viruses, monovalent inactivated oil-emulsion vaccines, containing either only GPV (LB strain) or MDPV (FM strain) whole-virus antigens, were prepared. The quantity of the GPV and MDPV whole-virus antigen incorporated in the monovalent vaccines was set to be equal to the amounts of antigens in the commercial bivalent vaccine, based on the humoral immune response induced when tested in SPF chickens (7.8 log_2_ VN titer against the LB strain of GPV, and 7.7 log_2_ against the FM strain of MDPV). The bivalent vaccine used in the study was a commercial inactivated oil-emulsion parvovirus vaccine (DEPARVAX^®^, CEVA-Phylaxia Veterinary Biologicals Co. Ltd., Budapest, Hungary). Each vaccine was applied by the sub-cutaneous (SC) route in a dose of 0.2 mL for the immunization of the ducklings.

### 2.3. Challenge Viruses

The challenge viruses used to evaluate the clinical protection provided by the tested vaccines were virulent field isolates of GPV (strain D17/99) and MDPV (strain FM). The challenge virus was administered intramuscularly into the tight muscle at a dose of 10^4^ EID_50_ for both the GPV and MDPV.

### 2.4. Virus Neutralisation Test

A virus neutralization (VN) test to measure the antibody titers to GPV and MDPV in the serum samples collected during the studies was done with the microneutralization method performed on primary goose embryo fibroblast cell cultures as described earlier [21,22] with some modifications. Briefly, the serum neutralization was carried out against 400 TCID_50_ of either GPV (strain LB) or MDPV (strain FM) and the neutralization titer was determined after a 6-day incubation period. The reading was carried out based on the detection of parvovirus antigen from the cell culture supernatants at the end of the incubation period by an in-house ELISA method [22].

### 2.5. Vaccination/Protection Experiments

Challenge experiments were carried out both in MDA-free and MDA-positive ducks. In the experiment on MDA-free birds, vaccination was done at one day of age, while in the experiment on MDA-positive birds a prime-boost vaccination regime was tested: primary vaccination at one day-old and booster vaccination at 3 weeks of age.

#### 2.5.1. Cross-Protection between GPV and MDPV in MDA-Free Muscovy Ducks

Ducklings at one day of age were assigned to groups balanced for weight. Ninety ducklings were allocated into 3 groups (Groups A, B and C); Groups A and B (vaccinated groups), 30 ducks each, received monovalent GPV or MDPV vaccine, respectively at one-day of age. Before the challenge at 15 days of age, each vaccinated group was divided into two subgroups, each of which containing 15 birds, then was moved to separate isolated animal rooms for the challenge either with GPV or MDPV. Thirty unvaccinated ducklings of Group C were divided into 3 subgroups: two subgroups served as positive controls for the challenge either with GPV or MDPV and one subgroup was a negative control (non-vaccinated, unchallenged). Challenges of the relevant groups either with GPV or MDPV were done at 15 days of age and the post-challenge observation period was 3 weeks. The design of the experiment is summarized in Table 1.

The protection levels obtained in the groups vaccinated with monovalent vaccines against the homologous and heterologous challenge virus were compared with each other. The basis of the evaluation of protection is described below under paragraph 2.6.

Blood samples were taken of the wing vein from all birds at the time of challenge (15 days of age) and all surviving ducks at the end of the post-challenge observation period (36 days of age). The changes in VN antibody titers to GPV and MDPV following challenge were determined both in the vaccinated and unvaccinated ducks at the end of the observation period (D36) and were compared to the VN titers measured at the time of challenge (D15).

#### 2.5.2. Immunogenicity of the Bivalent Vaccine in MDA-Free Muscovy Ducks

The outline of the study performed to monitor the humoral immune response to the bivalent vaccine, and the protection against the challenge, is detailed in Table 2.

Ducklings in the vaccinated group received one dose (0.2 mL) of the bivalent vaccine by subcutaneous route at one day of age. Challenge was done at 15 days of age and the post-challenge observation period was 3 weeks. The changes in VN antibody titers to GPV and MDPV following the challenge were checked both in the vaccinated and unvaccinated ducks at the end of the observation period (D36), and compared to the VN titers measured at the time of challenge (D15). Vaccinated unchallenged birds were sampled at the same ages and their results were used for calculating the relative titer increase in the vaccinated challenged birds (the results of VN titer increase between D15 and D36 in the vaccinated unchallenged group subtracted from the results of VN titer increase in vaccinated challenged groups). The vaccinated unchallenged group (Group A3) was kept until 64 days of age to follow the duration of antibody response after the day-old single vaccination.

#### 2.5.3. Immunogenicity of the Bivalent Vaccine in MDA-Positive Muscovy Ducks

The design of the experiment carried out to evaluate the protection against the challenge and to monitor the humoral immune response to the bivalent vaccine is outlined in Table 3.

The vaccinated ducklings along with the unvaccinated ones were assigned to subgroups (each subgroup consisted of 10 birds) for the challenge with one of the two viruses (GPV and MDPV) at different ages. The level of protection against GPV and MDPV challenges was first evaluated at 3 weeks of age for ducks that received only a single day-old vaccination. Further challenges were carried out at 4, 5, and 6 weeks of age for the birds that received the full prime-boost vaccination regime (primary vaccination at one day old and a booster at 3 weeks of age).

To monitor the antibody response to the vaccination and the decay of MDA, ducks in Group B (vaccinated control) and Group D (unvaccinated control) were sampled for serology at one day-old, then at 3, 4, 5, 6, and 9 weeks of age.

### 2.6. Evaluation of Protection

Protection was evaluated based on mortality, clinical signs indicative of parvovirus infection, and body weight gain (evaluated on individual bases) during a 3-week post-challenge observation period. The clinical signs or death were considered specific for parvovirus infection if at least one of the following criteria were met: (i) presence of typical clinical signs (leg weakness and/or diarrhea, with feathering disorders, and marked growth retardation), (ii) the presence of typical gross pathological changes in the case of mortality (ascites, hepatitis, myocardial degeneration, and myocarditis confirmed by histological examination), and (iii) the relative body weight gain was lower than the average of the controls minus twice the standard deviation. Since the severity and clinical presentation of the disease caused by WPVs is strongly influenced by the age of the birds as well as by the presence of residual MDA at the time of infection, the body weight gain was the most important indicator used for the evaluation of protection. To allow a better comparison of the growth rate of individual birds independently from their gender, the relative body weight gain (body weight at end of observation period divided by the body weight at challenge) was calculated instead of the absolute body weights. The birds showing any signs of the disease or that died were considered non-protected, while the ducklings free from the listed symptoms were considered protected. The protection level was calculated by dividing the number of protected birds by the number of tested animals in the group.

### 2.7. Statistical Analysis

Clinical protection and VN titer results were analyzed with Statgraphics Centurion XVI software (version 16.2.04), using an analysis of variance (ANOVA) to statistically test the equality of results in the compared groups. The confidence level of the test was 95% (*p* value < 0.05 indicates statistically significant difference). When a pair-wise comparison was made between two groups, a standard t-test was used to analyze statistical difference between the means of the two samples. *p*-values of different significance are represented by number of asterisks on graphs: *p* > 0.05 (no significant difference); 0.01 < *p* < 0.05 (*); 0.001 < *p* < 0.01 (**); 0.0001 < *p* < 0.001 (***); *p* < 0.0001 (****). Results of clinical protection were analyzed by the Fisher’s exact test at a 95% confidence level.

## 3. Results

### 3.1. Cross-Protection between GPV and MDPV in MDA-Free Muscovy Ducks

Monovalent vaccines provided 100% protection against challenge with the homologous virus, while conferring only partial protection against challenge with heterologous viruses. The challenge of monovalent vaccinated birds with the heterologous virus resulted in significantly lower protection (*p* < 0.001). The vaccine containing goose parvovirus antigen provided 26.7% protection against challenge with the heterologous MDPV and the one containing duck parvovirus antigen resulted in 13.3% protection against GPV challenge. (Table 4).

In animals immunized with the monovalent GPV vaccine, the challenge with the homologous virus boosted the antibody titer to GPV, while the antibody response measured against the heterologous MDPV remained significantly lower (*p* < 0.0001) (Figure 1). Opposite to this, challenge with the heterologous virus (MDPV) induced an antibody rise to both the homologous and heterologous viruses, which was not significantly different (*p* = 0.4370) (Figure 1).

On the other hand, in animals immunized with the monovalent MDPV vaccine, the challenge either with the homologous or heterologous virus induced significant antibody titer increase against both viruses, however the differences between the titer increase measured against the homologous and heterologous virus were less significant (*p* = 0.0323 and *p* = 0.0218 after GPV and MDPV challenge, respectively) than the ones obtained in the GPV-vaccinated birds (Figure 2).

In the non-vaccinated ducks, high VN titers were detected after a challenge with infection either with GPV or MDPV against the homologous viruses, but were much lower against the heterologous ones. The differences were highly significant (Figure 3).

### 3.2. Immunogenicity of the Bivalent Vaccine in MDA-Free Muscovy Ducks

Day-old vaccination of ducklings induced a low level of VN antibodies to GPV and MDPV by two weeks of age, however, there was a significant titer increase (*p* < 0.001) by 5 weeks post-vaccination against both viruses and these titers were maintained almost at the same level until the last sampling date at 64 days of age. The VN titer to MDPV reached a slightly higher level than the titer measured against GPV (Figure 4).

The results of clinical observation after the challenge with GPV and MDPV are presented in Table 5.

All vaccinated ducks remained healthy during the 3-week long post-challenge observation period based on all parameters used for the evaluation of protection. Unvaccinated ducks died or had characteristic clinical signs of parvovirus infection in 90% of GPV-infected and 100% of MDPV-infected birds.

The results of anti-parvoviral VN titers measured in the serum samples collected from the vaccinated and unvaccinated ducks at the time of challenge infection (D15) and at the end of the challenge experiment (D36) are summarized in Table 6 and shown on Figure 5 and Figure 6.

In the non-vaccinated animals, infection with any of the two viruses (GPV or MDPV) resulted in significantly higher (*p* < 0.001) VN antibody titers to the homologous virus than against the heterologous virus (Table 6 and Figure 5, subgroups C1 and C2) 3 weeks after infection (D36).

The increase of VN antibody titer to GPV following the challenge infection of vaccinated groups (group A1 and A2) was significantly bigger than the increase measured at the same time-interval (3 weeks) in the vaccinated non-challenge group (Table 6 and Figure 6, groups A1, A2, and A3). However, the relative titer increases, calculated by subtracting the values of titer increase measured between D15 and D36 against GPV and MDPV in the vaccinated unchallenged groups from those measured during the same period in the vaccinated and challenged groups, indicated a pronounced reduction of the booster effect, especially after the heterologous challenge (Table 6).

### 3.3. Immunogenicity of the Bivalent Vaccine in MDA-Positive Muscovy Ducks

The maternally derived antibody levels to GPV and MDPV in the commercial ducklings were moderately high at day-old. By 21 days of age, the VN antibody titers to both GPV and MDPV declined below 2 log_2_ in the non-vaccinated control birds, while in most of the vaccinated birds the decline was less pronounced and the VN titer levelled above 3 log_2_ (Figure 7). After the booster vaccination at 3 weeks of age, the antibody titer increased to both GPV and MDPV and reached high levels by 5 to 6 weeks of age and remained high until 9 weeks of age (end of the observation period), while the non-vaccinated birds remained serologically negative (Figure 7).

Protection against clinical disease following challenge of the unvaccinated birds decreased gradually with age due to the decline of MDAs. Clinical protection against GPV challenge was 100, 90, 70, and 60% at 3, 4, 5, and 6 weeks of age, respectively, while against MDPV it was 90, 80, 40, and 30%, respectively. In the vaccinated ducks, 100% protection was observed against both viruses at the same ages, except in the subgroup challenge with GPV at 6 weeks of age where 90% protection was measured.

## 4. Discussion

Although the molecular properties of WPVs have been well characterized, less is known about their antigenic structure. Using bacterially expressed VP1 proteins, seven antigenic regions of VP1 were identified that reacted with sera from a GPV-infected geese [23]. Through bioinformatic prediction, there were one epitope on non-structural protein and three epitopes on structural proteins between MDPV and GPV that might cross-react with each other. The four epitopes were expressed in *Escherichia coli* and found to react with GPV- and MDPV-antisera via Western blot [24]. However, no comprehensive mapping of the antigenic structure of GPV and MDPV has been done to identify the level of antigenic differences between the two viruses. Since the VP1 polypeptides of GPV and MDPV share about 76-80% identity at the nucleotide and 88% identity at the amino acid level, it has been suggested that there may be immunogenic cross reactivity between GPV and MDPV [2,7,23,24], however the level of this immunological cross-reactivity has not yet been determined and published. Therefore, the investigation of cross-reactivity between GPV and MDPV through immunization seemed to be important to provide support for the development of a vaccine that meet field requirement for the control of both MDPV and GPV infection in Muscovy duck.

Infection of susceptible (MDA-free) ducks with GPV or MDPV induced antibodies that neutralized the homologous virus at significantly higher titer than the heterologous one (*p* < 0.001). These findings clearly demonstrate that there must be substantial antigenic differences between the two viruses. However, when vaccinated birds were challenged, the infection provoked a booster effect not only on the homologous, but also on the heterologous antibody response, indicating that there are some antigenic epitopes that are shared between GPV and MDPV, as reported by Ju et al. (2011) and Li et al. (2013). The cross-protection studies carried out by us in maternal antibody-free, susceptible Muscovy ducklings provided further evidence on the antigenic difference of the two viruses. Vaccination with monovalent vaccine (containing either GPV or MDPV whole virus antigen) provided a significantly higher level of protection (*p* < 0.001) against challenge with homologous viruses when compared to the protection level obtained when the animals were challenged with the heterologous virus. Only the bivalent vaccine containing both goose and Muscovy duck whole-parvovirus antigens protected the birds fully against the two waterfowl parvoviruses that can cause disease in Muscovy ducks.

It seems that the antigenic relationship between GPV and MDPV is not enough to elicit an adequate cross-immunity between the two viruses. The increase of antibody titer in vaccinated animals following homologous virus challenge was significantly lower than after a heterologous virus infection. This observation suggests that immunization not only prevents the birds from the clinical disease after homologous virus challenge but also suppresses the replication of the challenge virus substantially, which may contribute to a better control of virus transmission among the vaccinated animals.

Our study showed that the inactivated water-in-oil emulsion waterfowl parvovirus vaccine was able to induce a reasonably fast immunity in maternal antibody-free birds applied at one day of age, providing a high level of clinical protection (100%) by 2 weeks after vaccination. This protection had already been achieved when the virus-neutralizing antibodies induced by the vaccination were still very low or barely detectable. Testing the bivalent vaccine in day-old Muscovy ducklings with maternally derived antibodies to GPV and MDPV showed that it could induce an active immune response in face of MDA, resulting in a high level of clinical protection (90–100%) by three weeks of age against challenge with any of the two waterfowl parvoviruses. Boosting the primary immune response with a second vaccination at 3 weeks of age induced fast and strong immune response which conferred strong immunity to the birds till the end of the susceptible age.

Both the serology results and the clinical protection data generated in our experiments indicated that protection against the clinical disease caused by waterfowl parvoviruses in Muscovy ducks can be achieved only by administering a bivalent parvovirus vaccine.

The results of the experimental challenges and field observation reported by scientists indicate that infection of susceptible Muscovy ducks with waterfowl parvoviruses can result in chronic disease with substantial growth retardation at least until 6 to 8 weeks of age that may cause significant economic losses (15, 18, 20). Therefore, to prevent these losses due to infection during this susceptible period of life, active immunity against the disease should be induced by vaccination as early in life as possible before the maternally derived antibodies decline to an un-protective level. This can be achieved by inactivated vaccines as reported by Takehara et al. (1995) who used a monovalent inactivated GPV vaccine to control the disease caused by GPV infection in Muscovy ducks. Our immunological cross reactivity studies using monovalent and bivalent vaccine preparations of GPV and MDPV demonstrated that for the achievement of protection against the two waterfowl parvovirus species that can cause disease in Muscovy ducks, a vaccine containing both GPV and MDPV antigens that induce neutralizing antibodies should be applied.

## 5. Conclusions

The results of our investigation demonstrate that there are significant antigenic differences between GPV and MDPV. It was shown that the antigenic relationship between GPV and MDPV is not enough to elicit an adequate cross-immunity between the two viruses, therefore the use of bivalent vaccine containing both goose and Muscovy duck parvovirus antigens is necessary to achieve adequate protection against the two waterfowl parvoviruses that can cause disease in Muscovy ducks. It was also demonstrated that inactivated bivalent waterfowl parvovirus vaccine was able to induce a reasonably fast immunity even when applied in face of MDA, and if day-old vaccination is followed by a booster vaccination the immunity lasted till the end of the susceptible age of the birds.

## Figures and Tables

**Figure 1 vaccines-10-01255-f001:**
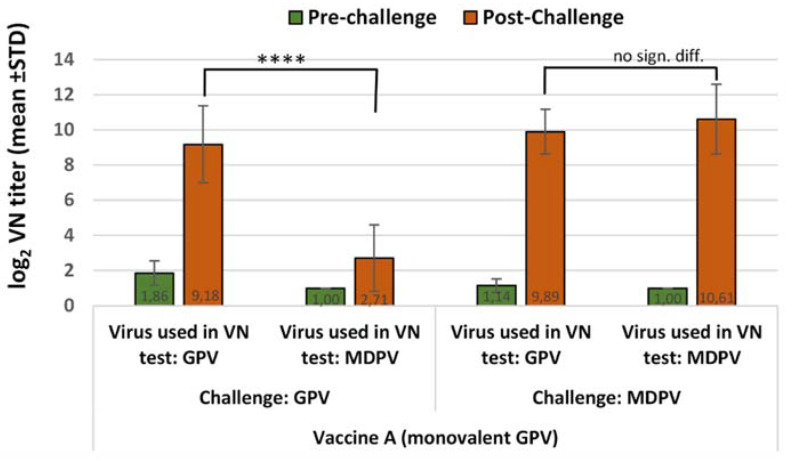
Humoral antibody response to GPV and MDPV in MDA-free ducks, vaccinated with monovalent GPV vaccine then challenged with homologous or heterologous strains of WPV at 2 weeks of age. Statistical comparison of post-challenge virus neutralization titers against GPV and MDPV indicate significant difference after GPV challenge (**** *p* < 0.0001), but no significant difference in the case of MDPV challenge infection (*p* = 0.437).

**Figure 2 vaccines-10-01255-f002:**
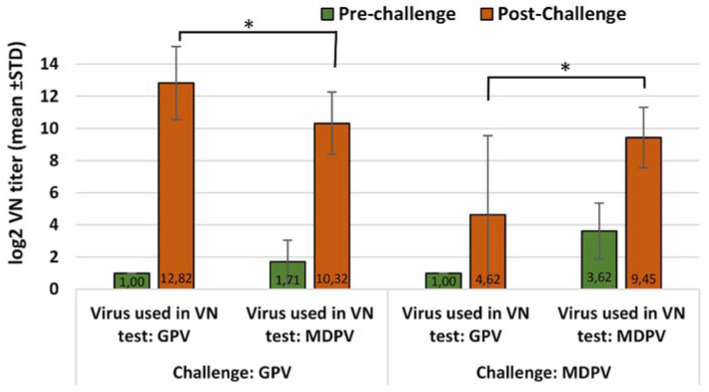
Humoral antibody response to GPV and MDPV in MDA-free ducks, vaccinated with monovalent MDPV vaccine then challenged with GPV or MDPV strains at 2 weeks of age. Statistical comparison of post-challenge virus neutralization titers against GPV and MDPV indicate significant differences in the case of both GPV and MDPV challenge infections (* *p* = 0.0323 and * *p* = 0.0218 respectively).

**Figure 3 vaccines-10-01255-f003:**
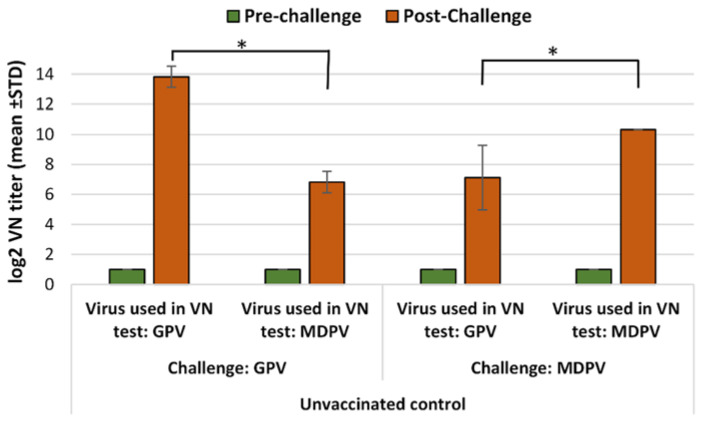
Humoral antibody response to GPV and MDPV in unvaccinated MDA-free ducks, following infection with GPV or MDPV strains at 2 weeks of age. Statistical comparison of post-challenge virus neutralization titers against GPV and MDPV indicate significantly higher titers against the homologous neutralizing virus than against the heterologous one (* in case of GPV infection *p* = 0.0105; in case of MDPV infection *p* = 0.0108).

**Figure 4 vaccines-10-01255-f004:**
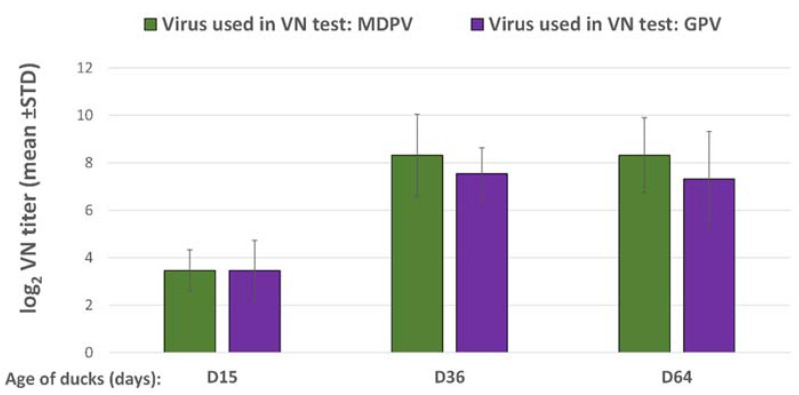
Antibody response of MD-free Muscovy ducks to the bivalent vaccine (log_2_ VN titer).

**Figure 5 vaccines-10-01255-f005:**
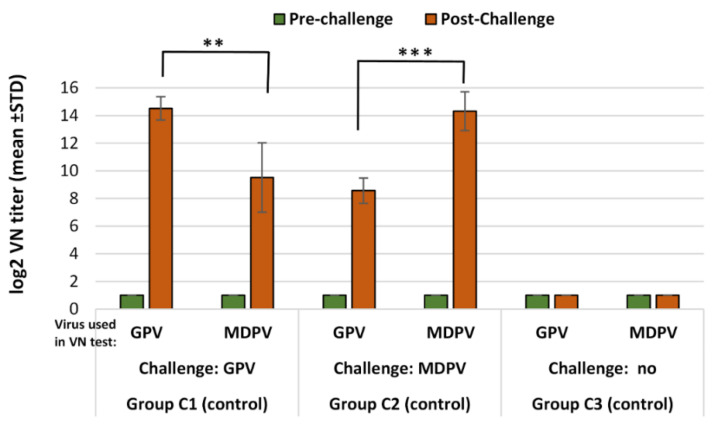
Development of antibody response to GPV and MDPV in the non-vaccinated MDA-free ducks, following challenge infection with GPV or MDPV at 2 weeks of age. Statistical comparison of post-challenge virus neutralization titers against GPV and MDPV indicate that significantly higher VN titers against homologous virus than against the heterologous one (in case of GPV infection ** *p* = 0.0022; in case of MDPV infection *** *p* = 0.0005). No zero-conversion was found in non-challenged birds.

**Figure 6 vaccines-10-01255-f006:**
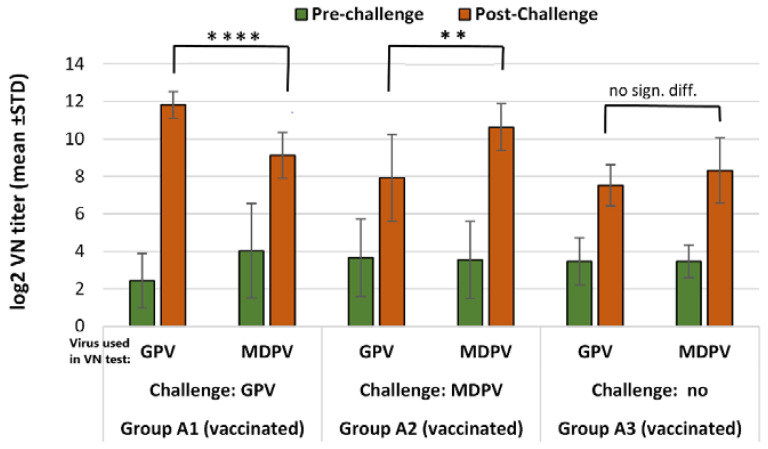
Humoral antibody response to GPV and MDPV in MDA-free ducks vaccinated with the bivalent vaccine, following challenge infection with GPV or MDPV strains at 2 weeks of age. The non-challenged vaccinated group (A3) was sampled at the same date when post-challenge samples were collected in groups A1 and A2, to determine the level of vaccine induced VN titers at the same time. Statistical comparison of post-challenge virus neutralization titers against GPV and MDPV indicated significantly higher titer increases against the virus homologous with the challenge virus (in case of GPV infection **** *p* < 0.0001; in case of MDPV infection ** *p* = 0.0045).

**Figure 7 vaccines-10-01255-f007:**
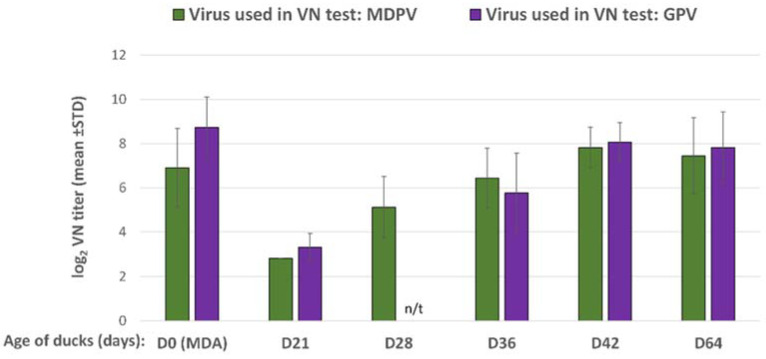
Humoral antibody response in MDA-positive ducks, vaccinated with the bivalent vaccine in a prime–boost regime. (n/t = not tested).

**Table 1 vaccines-10-01255-t001:** Cross-protection experiment in MDA-free Muscovy ducks.

Groups	Vaccine	Subgroups	No. of Ducks	Challenge Virus	Blood Sampling
A	Monovalent GPV	A/1	15	GPV	D15, D36
		A/2	15	MDPV	D15, D36
B	Monovalent MDPV	B/1	15	GPV	D15, D36
		B/2	15	MDPV	D15, D36
C Control	ND	C/1	10	GPV	D15, D36
		C/2	10	MDPV	D15, D36
		C/3	10	ND	D15, D36

Note: ND = not done.

**Table 2 vaccines-10-01255-t002:** Study design to evaluate the immunogenicity of bivalent vaccine in MDA-free Muscovy ducks.

Groups	Subgroups	No. of Birds	Challenge Virus	Blood Sampling
**A** **(vaccinated)**	A/1	10	GPV	D15, D36
	A/2	10	MDPV	D15, D36
	A/3	10	ND	D15, D36, D64
**C** **(Control, unvaccinated)**	C1	10	GPV	D15, D36
	C2	10	MDPV	D15, D36
	C3	5	ND	D15, D36, D64

Note: ND = not done.

**Table 3 vaccines-10-01255-t003:** Study design to evaluate the immunogenicity of the bivalent vaccine in MDA-positive Muscovy ducks.

Groups	Subgroups	Challenge Virus	Age at Challenge * (at Weeks)	Blood Sampling
**A** (vaccinated-challenged)	A1/1-4	GPV	3, 4, 5, 6	ND
	A2/1-4	MDPV	3, 4, 5, 6	
**B** (vaccinated control)	N/A	ND	N/A	D0 **, D21, D28, D36, D42, D64
**C** (unvaccinated-challenged)	C1/1-4	GPV	3, 4, 5, 6	ND
	C2/1-4	MDPV		
**D** (unvaccinated control)	N/A	ND	ND	D0 **, D21, D28, D36, D42, D64

Notes: * At each challenge date 10 ducks were challenged from each subgroup for each challenge virus. ** The age of the birds in days. At each sampling date 10 ducks were blood sampled. N/A = not applicable, ND = not done.

**Table 4 vaccines-10-01255-t004:** Cross-protection between GPV and MDPV in MDA-free Muscovy ducks: Clinical protection against challenge.

Groups	Vaccine	Challenge Virus	Morbidity * (%)	Mortality (%)	Protection (%)
A/1	Monovalent GPV	GPV	0	0	100
A/2	Monovalent GPV	MDPV	73.3	0	26.7
B/1	Monovalent MDPV	GPV	86.7	0	13.3
B/2	Monovalent MDPV	MDPV	0	0	100
C/1 (+ve control)	None	GPV	100	60	0
C/2 (+ve control)	None	MDPV	100	0	0
C/3 (−ve control)	None	None	0	0	N/A

* A duck was considered clinically affected if a typical feathering disorder and/or marked growth retardation, indicated by a significantly lower relative body weight gain, could be observed. N/A—not applicable.

**Table 5 vaccines-10-01255-t005:** Efficacy of bivalent vaccine in MDA-free Muscovy ducks—Protection against challenge.

Subgroups	Challenge Virus	No. of Birds with Clinical Signs/no. of Birds Tested	No. of Birds Died/no. of Birds Tested	Clinical Protection * (%)
**A1**	GPV	0/10	0/10	100
**A2**	MDPV	0/10	0/10	100
**C1**	GPV	9/10	3/10	10
**C2**	MDPV	10/10	6/10	0
**C3**	No	None	None	N/A

* Clinical protection expressed in the percentage of ducks that survived the 21-day clinical observation period without showing clinical signs or having significant growth retardation indicative of parvovirus infection.

**Table 6 vaccines-10-01255-t006:** Antibody response of vaccinated and non-vaccinated MDA-free Muscovy ducks to challenge infection (log_2_ VN titer).

Age at Sampling		D15 (Pre-Challenge)		D36 (Post-Challenge)		Relative Titer Increase **	
Virus Used in VN Test		GPV	MDPV	GPV	MDPV	GPV	MDPV
Subgroups *	Challenge Virus	Log_2_ VN Titer (Mean ± SD)					
A1	GPV	2.43 ± 1.45	4.03 ± 2.52	11.82 ± 0.71	9.12 ± 1.23	5.33	0.23
A2	MDPV	3.66 ± 2.06	3.56 ± 2.05	7.92 ± 2.32	10.62 ± 1.25	0.20	2.20
A3	No	3.46 ± 1.26	3.46 ± 0.87	7.52 ± 1.09	8.32 ± 1.73	N/A	N/A
C1	GPV	<1.00	<1.00	14.52 ± 0.84	9.52 ± 2.50	N/A	N/A
C2	MDPV	<1.00	<1.00	8.57 ± 0.96	14.32 ± 1.41	N/A	N/A
C3	No	<1.00	<1.00	<1.00	<1.00 *		

Note: * Subgroups A1–A3 vaccinated with bivalent vaccine, subgroups C1–C3 non-vaccinated controls; ** Relative titer increase was calculated by subtracting the values of titer increase measured between D15 and D36 against GPV or MDPV in the vaccinated unchallenged groups from those measured during the same period in the vaccinated and challenged groups. N/A—not applicable.

## Data Availability

The data presented in this study are available on request from the corresponding author. The data are not publicly available due to the company data management policy.

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
