# Peer review of "Immunogenic Cross-Reactivity between Goose and Muscovy Duck Parvoviruses: Evaluation of Cross-Protection Provided by Mono- or Bivalent Vaccine"

_vaccines, 2022, doi:10.3390/vaccines10081255_

Round 1

Reviewer 1 Report

The manuscript by Palya et al. describes a study performed to understand cross-protection and immunogenicity of antibodies elicited by immunization of Moskovy ducks with GPV and MDPV monovalent or bivalent vaccines. Overall the data suggest that immunization with both vaccines (either a combination of monovalent or a bivalent) is needed for full protection. While the novelty of the study is not clear, the study brings an incremental improvement in understanding of the topic. While the dataset seems to be reasonable, its presentation and description are lacking clarity, logic, and purpose. The specific comments are listed below:

Lines 12-13, Abstract - the sentence is unclear, I assume the authors meant that some birds were immunized with the inactivated GPV, others - with the inactivated MDPV, and yet others with both.

Line 16, Abstract - confusing again, were there antigens or inactivated virus particles?

Lines 17-20, Abstract - the sentence is unclear

Line 19, Abstract - spell out VN

Line 80, Introduction - the authors cite “substantial genetic differences between GPV and MDPV”, yet earlier they reported that these viruses have 88 - 98% identity at the amino acid level, which this reviewer would characterize as very similar and not substantially different

Line 113 - how the vaccines were inactivated?

Lines 118-121 - were the doses of each virus in the commercial bivalent vaccine equivalent to the doses of the virus in each monovalent vaccine?

Tables - use the same designation in all tables, for instance in Table 2 the authors are using “no” and in Table 3 - “ND”.

2.6 Evaluation of protection - 2 issues here: (1) the authors are not really evaluating protection but rather they are evaluating clinical symptoms and weight changes, thus the sub-heading for this paragraph should reflect that; (2) did the authors use some sort of scoring system? If yes, it needs to be presented, if no, how do they quantify the symptoms reported in Table 4?

Figure 1 - 2 issues here as well: (1) the legend for X axis is confusing, it says “virus used in VN test” but then lists antigens, this should be corrected; (2) when presenting statistical significance, the convention calls for using an increasing number of asterisks with an increase in significance - p<0.05 (*), p<0.01 (**), p<0.01 (***), and p<0.001 (****) (this comment applies to all figures)

Figure 2 - please verify your calculations regarding statistical significance, the error bars are grossly overlapping and it’s hard to believe that the 2 sets differ significantly. The same goes for Figure 4.

Why the results are presented in different formats (Figs 1, 2, 3 vs Tables 5 and 7)? This makes them harder to compare.

The significance of Table 7 is unclear, what point do the authors want to make? It is also unnecessarily complicated.

Figure 6 - several issues: (1) why the format is so different? (2) what do “Antigen: MDPV” and “Antigen: GPV” mean? Are these viruses used in VN? If yes, what does it have to do with antigen? (3) what happened with GPV result on day 28?

The discussion is really a summary of data with some additional information that belongs to the Introduction. The authors provide a conclusion, but its connection to the study is not very clear. The discussion needs to be re-written to discuss the data from this study in the context of knowledge available in the field.

Throughout the manuscript, the authors use interchangeably the terms “inactivated vaccine(s)” and “antigens”. Inactivated vaccines (sometimes called “killed vaccines”) are typically whole virions treated by various means to neutralize their ability to replicate (formaldehyde, radiation, etc). Antigenic or subunit vaccines are typically individual viral proteins expressed in a heterologous system and delivered in an adjuvant formulation. Therefore, using these terms interchangeably makes it confusing to a reader and needs to be corrected.

For all figures - the figure legends poorly describe the context of the figure leaving readers guessing what they are looking at.

This reviewer would also recommend seeking the service of a professional editor to review this manuscript for clarity.

Reviewer 2 Report

The authors investigated the immunogenic cross reactivity between goose GPV and MDPV, the results showed that the bivalent vaccine protected the birds (90-100%) against both parvoviruses that can cause disease in Muscovy ducks. However, the protection of monovalent vaccine against herterologous virus challenge was not good. The results is helpful for the clinic, but there are some questions shold be adressed.

1. It is easy to  understand that the vaccines work well against homologous virus challenge, why the authors don't choose other diffrent virus strains except LB and FM strains  to be tested? it maybe better to know the function of the bivalent vaccine.

2. The authors shoud provide more information about LB and FM strains, are they most popular in the clinic?

3.  Are there any adjuvants used in the inactivated vaccines? If yes, what are they?

4. "MDA-free" should be given the full name for the first time.

Author Response

Dear Reviewer,

Your comments and recommendations were highly appreciated by the authors. Please find below our answers in red to your.

The authors investigated the immunogenic cross reactivity between goose GPV and MDPV, the results showed that the bivalent vaccine protected the birds (90-100%) against both parvoviruses that can cause disease in Muscovy ducks. However, the protection of monovalent vaccine against herterologous virus challenge was not good. The results is helpful for the clinic, but there are some questions shold be adressed.

  1. It is easy to  understand that the vaccines work well against homologous virus challenge, why the authors don't choose other diffrent virus strains except LB and FM strains  to be tested? it maybe better to know the function of the bivalent vaccine.

Since GPV and MDPV pose low mutation rate and within the GPV and MDPV groups the differences are only about 0.1–7.0% and 0.1–1.9%, respectively, we think that using well characterized strain of GPV and MDPV would represent, in antigenic nature, other circulating field virus strains belonging to the same virus group.

  1. The authors shoud provide more information about LB and FM strains, are they most popular in the clinic?

These strains are among the representative isolates of the two virus groups, well characterized and described in the literatures. (Derzsy, D.: A viral disease of goslings. Acta Veterinaria Hungarica 1967, 17, 443-48.; Zádori, Z.; Stefancsik, R.; Rauch, T.; Kisary, J.: Analysis of the complete nucleotide sequences of goose and Muscovy duck parvoviruses indicates common ancestral origin with adeno-associated virus 2. Virology 1995, 212, 562–573.; Glávits, R.; Zolnai, A.; Szabó, É.; Zarka, P.; Mató, T.; Palya, V.: Comparative pathological studies on domestic geese (Anser anser domestica) and Muscovy ducks (Cairina moschata) experimentally infected with parvovirus strains of goose and Muscovy duck origin. Acta Veterinaria Hungarica 2005, 53, 73-89.)

  1. Are there any adjuvants used in the inactivated vaccines? If yes, what are they?

The information on the vaccine formulation is given in the revised version of the paper (line 122-124)

  1. "MDA-free" should be given the full name for the first time.

Done.

26 July, 2022.

Sincerely,

Vilmos Palya

Round 2

Reviewer 1 Report

No additional comments

Reviewer 2 Report

I have no other suggestions.